# Key Challenges and Emerging Technologies in Industrial IoT Architectures: A Review

**DOI:** 10.3390/s22155836

**Published:** 2022-08-04

**Authors:** Akseer Ali Mirani, Gustavo Velasco-Hernandez, Anshul Awasthi, Joseph Walsh

**Affiliations:** 1IMaR Research Centre, Munster Technological University, V92 CX88 Tralee, Ireland; 2CONFIRM Research Centre, Unit 2 Park Point, Dublin Road, Castletroy, V94 C928 Limerick, Ireland; 3Lero, The Irish Software Research Centre, Tierney Building, University of Limerick, V94 NYD3 Limerick, Ireland

**Keywords:** blockchain, edge/fog computing, IIoT architectures, Industry 4.0, interoperability, low latency, reliability, scalability, security, software-defined networking

## Abstract

The Industrial Internet of Things (IIoT) is bringing evolution with remote monitoring, intelligent analytics, and control of industrial processes. However, as the industrial world is currently in its initial stage of adopting full-stack development solutions with IIoT, there is a need to address the arising challenges. In this regard, researchers have proposed IIoT architectures based on different architectural layers and emerging technologies for the end-to-end integration of IIoT systems. In this paper, we review and compare three widely accepted IIoT reference architectures and present a state-of-the-art review of conceptual and experimental IIoT architectures from the literature. We identified scalability, interoperability, security, privacy, reliability, and low latency as the main IIoT architectural requirements and detailed how the current architectures address these challenges by using emerging technologies such as edge/fog computing, blockchain, SDN, 5G, Machine Learning, and Wireless Sensor Networks (WSN). Finally, we discuss the relation between the current challenges and emergent technologies and present some opportunities and directions for future research work.

## 1. Introduction

The Internet of Things (IoT) has brought a revolution in the current century by enabling ubiquitous and exponential connectivity of billions of devices and accessing them from any place at any time [1]. The initial concept of IoT as the connection of real-world objects with the internet was given by Kevin Ashton in 1999 [2], and the International Telecommunication Unit (ITU) further extended it to be the connection between people and things and between the physical and virtual objects for the exchange of information to perform coordinated tasks [3]. As the IoT is achieving smart objectives without human involvement by connecting real-world applications [4], Industrial IoT is further bringing an evolution in the manufacturing process by withstanding the mission-critical requirements as compared to IoT [5]. IIoT is helping industries to increase operational efficiency with the convergence of Information Technology (IT) and Operational Technology (OT) [6]. Moreover, the new era of the Fourth Industrial Revolution (Industry 4.0) is further bringing paradigm shifts with the integration of IIoT and Cyber-Physical Systems (CPS) to provide insights for the collaborative work of intelligent devices [7]. While Industry 4.0 applies to any industry to provide self-optimization, better decisions from advanced sensors, production quality, and predictive maintenance for minimizing the system downtime [8], CPS integrates the networking, sensing, and computational features with physical systems to learn and adapt themselves [9]. The convergence of IoT, IIoT, and CPS forms the Industry 4.0 component that is achieving new heights in the current industrial revolution.

The digitization of industrial processes requires new technological advancements, which imposes big challenges for the end-to-end integration of IIoT. While the IoT has revolutionized the world by connecting anything via the internet, IIoT is further bringing an evolution in industries by addressing the stringent requirements compared to IoT. Figure 1 shows how IoT and IIoT differ in terms of target applications and requirements. For the end-to-end development of IIoT systems, different reference architectures such as Reference Architectural Model Industrie 4.0 (RAMI 4.0) [10], Industrial Internet Reference Architecture (IIRA) [11], and OpenFog Reference Architecture [12] provide the blueprint guidelines containing the set of architectural layers from sensors to the enterprise management features. Moreover, IIoT architectures composed of different layers are present in the literature to address the challenges and for the flexible integration, management, and control of collaborative services [13]. While the reference architectures provide the general layout for the development process without any fixed support of protocols and standards [12], the IIoT architectures present in the literature address the specific challenges, either in a particular use case in industry or the general-purpose industrial use. The proposed solutions in the literature share some attributes of using emergent technologies to develop layered architectures.

Different reviews and surveys on IIoT architectures are present in the literature focusing on researching specific topics in terms of technologies and challenges. Some IIoT reference architectural studies are available in [16,17,18,19], where the authors have compared them, highlighted their limitations, and provided the findings for the best practices and solutions. Hazra et al. in [20] also review reference architectures and present a state-of-the-art survey on standards, protocols, and technologies addressing the interoperability issues in IIoT. The researchers in [21,22,23,24,25,26] review the proposed IIoT architectures, provide insights on how they are addressing the challenges, and highlight the features of some emerging technologies. The related works in the literature lack in classifying the relationship between the main IIoT requirements and the emerging technologies and providing transitional information on why there is even a need for the proposed architectures in the presence of reference architectures. Table 1 shows the related review and survey papers in literature.

In this paper, we provide a state-of-the-art review of IIoT architectures, comparing some widely accepted IIoT reference architectures and detailing recently proposed architectures in literature. The three main aspects of this review are that we highlight key challenges in adopting the IIoT architectures, identify the use of emergent technologies in IIoT systems, and analyze the role of these technologies in addressing those challenges. The rest of the paper is structured as follows: Section 2 presents the review and comparison of RAMI 4.0, IIRA, and OpenFog reference architectures. In Section 3, we identify the main IIoT requirements for its end-to-end development from the factory floor to the enterprise services, the emerging technologies used in IIoT architectural papers for presenting the solutions and addressing the challenges, and current research on IIoT architectures. Section 4 identifies how the conceptual and experimental architectures address these challenges, the relation of emerging technologies to IIoT requirements, and the scope of literature in addressing these requirements and using the emerging technologies. In Section 5, we summarize the findings and identify the potential research directions to address the challenges.

## 2. Industrial IoT Reference Architectures

A reference architecture provides the minimum functional requirements for a common ground to develop and analyze the systems [28]. The reference architectures in IIoT are independent of specific technologies and standards [12]. It provides the structural guidelines for multiple aspects of a system, including the standard networking model for the interaction with devices and sensors. It also provides the cloud architecture services for the remote monitoring and management features and the information on what hardware components the architecture support [29]. Experts from different organizations have proposed reference architectures to provide the necessary structure and transform the manufacturing process in industries based on the available technologies [16]. Three of the main IIoT reference architectures are RAMI 4.0, IIRA, and OpenFog RA, which are detailed below.

### 2.1. Reference Architectural Model Industrie 4.0 (RAMI 4.0)

Reference Architecture Model for Industrie 4.0 (RAMI 4.0) was developed in Germany to modernize the manufacturing process and industrial automation with the standardization of DIN SPEC 91345:2016 and IEC/PAS 63088:2017 [30]. In Industry 3.0, the products are isolated from each other, functions are bound to hardware, and system components interact across hierarchy levels. According to RAMI 4.0 RA information for Industry 4.0, the products are part of the network, functions are distributed throughout the network structure, and the participants can communicate with each other irrespective of the system hierarchy [31]. Figure 2 highlights how the RAMI 4.0 distinguishes Industry 4.0 from Industry 3.0.

In RAMI 4.0, the international standards for electronics, electrical, mechanics, and Information Technology (IT) participate in interdisciplinary ways to deploy the technology. It is based on Service-Oriented Architecture (SOA) for provisioning services between system components through network protocols and converting complex tasks into easy processes based on independent technologies and products [33]. Figure 3 shows the three-dimensional RAMI 4.0 RA model that provides insights into the framework where all the industrial partners can interact and understand each other and know how to adopt industry 4.0 in a structured way.

#### 2.1.1. Hierarchy Levels Axis

The hierarchy levels on the right horizontal axis of the model are on the IEC 62264 and IEC 61512 international standards for Information Technology (IT) and Control Systems (CS). The terms Station, Work Centers, Enterprise, and Connected World are included in the hierarchy axis from these standards for the common ground of current factory automation and process industry sectors [10]. Based on [34,35], the following are the seven levels in the hierarchy axis of the RAMI 4.0 model:*Product:* The product is the final outcome of the manufacturing in industry.*Field Device:* These are the hardware components such as sensors and actuators that collect the environment values.*Control device:* Controlling devices such as PLCs and DCs take the readings from sensors and send the controlling commands to operate the system.*Station:* This is the place where the user with administrative rights monitors the industrial activity and takes care of processes and events, e.g., SCADA.*Work Centers:* This provides the data storage, information, and analysis (MES) based on the historical insights.*Enterprise:* The enterprise level is followed (ERP) to manage all information and carry-out business profitable decisions. It keeps track of production vs. orders, expenses vs. revenue, and manages the manufacturing planning.*Connected World:* The system is connected to the internet to remain connected with the supply-chain process with external industries.

#### 2.1.2. Life Cycle Value Stream

The life cycle process standards used in Industrial automation, control, and measurement systems are on the left horizontal axis of the RAMI 4.0 model. The process shows the information of manufacturing components from the designing stage to the complete product. The Type field is related to the Design and Prototype level of manufacturing, while the Instance field is related to when the product is finally manufactured [10,36].

#### 2.1.3. Architecture Layers of RAMI 4.0 Model

The vertical layers are also called interoperability layers, which cover all the industrial processes, from the physical devices and assets to the integration of humans, technology, and protocols, along with the functional properties of system components and their business processes [33,36]. The researchers in [34,37,38,39] explain the following architectural layers of the RAMI 4.0 model:*Asset:* This is the lowest layer, which contains all the physical components, including the devices and peripherals.*Integration:* This layer provides the information generated from assets to the upper layers, enables the command and control of assets to the application and functional layer, and contains the IT elements such as RFID, HMI, and actuators.*Communication:* This layer is responsible for maintaining the communication between networks using the standards and protocols and enables the interaction of asset and Integration layers with the upper layers.*Information:* This layer provides the pre-processing of information for different events, as well as ensures sure the integrity and quality of data received from the lower layers, and then presents the structured data to the Functional and Business layers.*Functional:* The functional layer receives the data from the Assets layer and carries out the decisions based on data analytics.*Business:* This layer covers the enterprise business models and legal frameworks along with the industrial real-time monitoring services using the dashboards and user interaction applications.

### 2.2. Industrial Internet Reference Architecture (IIRA)

International Industrial Consortium (IIC) provides a common framework architectural model IIRA to address the support of diverse applications and standards for developing IIoT solutions. The IIRA is adapted based on the ISO/IEEE/IEC 42010 standards, and it can address the change in industrial control systems in the following ways [11]:*Increasing local collaborative autonomy:* It includes the provision of new technologies, computational power, and improved sensing, which will provide enhanced data accuracy and further assist in creating autonomous systems.*Increasing system optimization through global orchestration:* It includes data analytics using machine learning on collected sensor data to provide insights about the deployed system for system optimization and enhanced control systems.

IIRA is a three-tier system architecture containing an Edge Tier, Platform Tier, and Enterprise Tier. Different nodes, devices, sensors, control systems, and assets connected to the Edge Gateway via wireless and wired connections form a Proximity Network. The Edge Gateway performs the Device Management and Aggregation, then sends the relevant data to the Platform Tier via the Access Network. The Platform Tier performs the data transformation, operations, and analytics; and then sends the information to the Enterprise Tier via the Service Network. On Enterprise Tier, the user performs the monitoring and controlling under the Domain Applications and sends the controlling commands back to the Platform Tier through the Control Flow process. The Platform Tier then sends this information to the Edge Tier to perform the relevant tasks. Figure 4 shows the three-tier IIoT architecture given by IIC.

#### 2.2.1. Functional Domains and Functional Viewpoints

IIRA contains two important functional parts in its architecture, the Functional Viewpoint, and Functional Domain. The Functional Viewpoint is the overall architectural view of system components and their structure. The Functional Domain contains five distinct domains, which are the building blocks of the system architecture. Figure 5 highlights the information process between the functional domains of the IIRA model. The green arrows show the Data/Information Flows, the grey/white arrows show the Decision Flows, and the red arrows show Command/Request Flows.

#### 2.2.2. Functional Domains

*Control Domain:* It contains the functions for implementing the control systems in industries. It includes the sensing and actuation functions, which read the data from sensors and carry out the controlling signals for the actuators. It also contains the communication function that enables the information exchange between the system components and technologies using different features such as APIs. The control domain also interprets the system behavior and conditions by using modeling on the sensors’ data.*Operations Domain:* It carries outs the management and operation tasks for the control domain. It also provides the Provisioning and Deployment functions to access the assets remotely on a large scale and track, add, modify, or remove them regardless of the harsh industrial environment.*Information Domain:* This functional domain handles the data processing and collection from system components and performs the data analytics to acquire information about the system parameters and optimize the system through the decision-making steps.*Application Domain:* The Application Domain contains functions for implementing the application logic and rules for high-level optimization. It also includes the APIs and UI by which the relevant information is available for human interactions or different applications for processing.*Business Domain:* It contains different functionalities to support the business activities and processes and integrate them into the IIoT systems. Examples of the business functionalities are ERP, MES, Payments, and Billings, etc.

### 2.3. OpenFog Reference Architecture

This architecture facilitates the researchers, developers, designers, and industries to make needed components for fog computing. OpenFog provides the Fog as a Service (FaaS)-based architectural model to address industrial implementation issues through its compatibility with SaaS, PaaS, and IaaS. The OpenFog RA has many applications in industries, including smart vehicles and traffic control systems, smart cities, smart buildings, etc. It aims to provide security, cognition, agility, low latency, and efficiency. Moreover, the OpenFog RA is formed based on eight main pillars representing the overall system model attributes for the real-time deployments. The Perspective highlights the cross-cutting features of RA, while the View represents the structural aspects of the layered architecture. The View component contains three stakeholder views in the RA as Software View, System View, and Node View [12]. Figure 6 shows the OpenFog RA model. The light green colored vertical layers are the perspectives of RA, the light yellow and blue colored layers highlight the Node View and Software Architecture View, and the layers under the red border line show the System Architecture View.

#### 2.3.1. Eight Pillars of Fog Computing Architecture

The OpenFog RA is formed based on the core principles of eight pillars. These pillars represent the main attribute’s deployed systems manifest as per the given layered RA and fog computing technology.

*Security Pillar:* The security of OpenFog architecture is not just limited to the specific standards; it also contains all the mechanisms for security from the hardware component to the software-based application level. The security attributes presented in the OpenFog RA are data privacy and integrity, anonymity, attestation, measurement, trust, and user and device verifications. The OpenFog model provides end-to-end security. Moreover, the network link is provided between the nodes after information attestation is completed, followed by the verification process.*Scalability pillar:* This model provides the features where the individual fog nodes, storage services, and networks can scale based on the users’ requirements. There are the following scalability types in the OpenFog RA:Scalable performance: It includes improved fog performance as per the application demands by reducing the latencies in the system.Scalable capacity: It helps increase the network, system, applications, and user capacity.Scalable reliability: Scalable reliability is ensured by adopting the redundant fogs when there is a network fault or overload of information or processing.Scalable security: It includes the additional software and hardware security features such as access provision and crypto-based information processing when the security is becoming stringent.Scalable hardware: It enables the provision of additional hardware components upon requirement between the fogs in network and their internal systems, such as data storage, scale of wired and wireless networks, and the scaling of computational processes.*Openness pillar:* This pillar supports the diverse environment where the fog nodes and devices form an interoperable network by removing the negative impacts such as the quality and cost of a single vendor. It enables open communication between the components with location transparency and interoperability.*Autonomy pillar:* The autonomous structure avoids centralized processing by providing the decision-making facility near the devices for efficient operations, security, and cost. It enables the network discovery option, which allows the devices to keep alive if there is an uplink connection problem.*Programmability pillar:* The programming of the deployed nodes and system is available at the hardware and software layers with the ability to re-tasking the fog node. The programmability provides optimized security with automatic security patch updates, along with adaptable infrastructure and multi-tenancy.*Reliability, Availability, and Serviceability (RAS) pillar:* While reliability ensures the fog nodes and overall system components are working to deliver their functionalities under the given conditions, the availability functionality refers to the continuous management and back-end support, including the redundant and secure access from devices and redundant configurations. The serviceability enables the automated installation, up-gradation, and maintenance of fog nodes by supporting easily swappable hardware components.*Agility pillar:* This pillar is responsible for dealing with the changes occurring in the system and providing analytical insights from the extensive data received from the sensors to carry out efficient business decisions.*Hierarchy pillar:* Although in OpenFog RA, not all the systems are hierarchy-based, this pillar provides complementary and traditional hierarchy-based information for the enterprise systems.

#### 2.3.2. Perspectives

The Perspectives shown in the vertical green columns in Figure 6 are described below:*Performance and Scale:* The performance of deployed systems is under continuous care for the Quality of Service (QoS) and low latency by using time-sensitive networking and critical computing. The measurement of throughput and latency of a fog node defines the performance of fog computing that can be improved by bringing the fog computing closer to the edge. The new virtualization and containerization technologies in fog computing further improves the nodes’ scalability and isolation. These technologies can also carry out priority-based network traffic and resource allocation.*Security:* The fog architecture is not secured until trustworthiness is absent between the system components. The fog node hardware is secured with appropriate measures, and the complete data security and integrity are ensured from the low-level hardware to the software level with end-to-end security encryption. The security perspective also contains threat detection and privacy preservation features.*Manageability:* The manageability perspective provides the capability of responding and making decisions similarly to humans with the help of machine algorithms. It enables efficient manageability functions for a wide range of actions compared to the traditional IT and OT systems. Furthermore, it takes care of all the management functions, including the system alerting, operation and maintenance, the discovery of devices and nodes, etc.*Data, Analytics, and Control:* As the industries are generating high data for performing the analytics to make decisions, the traditional analytics approach is suitable for the increasing demands. Moreover, as the companies are moving forward to predictive maintenance from monitoring the system parameters, it is difficult to face the stringent requirements. Fog computing helps achieve these objectives by performing the data analytics at the edge closer to the source for specific analysis and sending the relevant information to the cloud services for business operations and business-related processing.*IT Business and Cross Fog Applications:* It highlights that fog applications need to operate at any hierarchical level and share the data with other nodes, ensuring the data interoperability to maximize the values from IT Business perspectives in a multi-vendor nature.

#### 2.3.3. Node View

It’s the lowest level view used in the OpenFog RA. The light yellow colored layers in Figure 6 highlight the Node View aspects of the architecture. These are necessary aspects to address before adding a node into the fog computing network.

*Node Security:* The Node Security represents both the vertical security perspectives and horizontal layer requirements as system security is critical from the silicon to the software level.*Node Management:* It supports the system management process by enabling management interfaces from the nodes. These interfaces support the monitoring and controlling of low-level nodes from high-level management systems.*Network:* The network part enables the nodes to communicate and share the information within the network based on the time-sensitive and time-aware networking.*Accelerators:* The accelerators used in fog applications improve the power and communication latency depending on the network scenario.*Compute:* The fog nodes run the open-source software at their node level for the basic computation and the interoperability between other nodes and system components.*Storage:* As it is necessary for a node to store data before learning or performing analysis, it requires a reliable storage device that should perform well with data integrity requirements and inform the storage device’s health condition.*Sensors, Actuators, and Control:* These are the lowest level architectural elements of an IoT system. While some of these devices have processing capabilities, some are dumb and cannot process the data. These elements are connected to the system by using the wired or wireless connection.*Protocol Abstraction Layer:* This layer is responsible for interfacing the sensors and actuators with the fog node for performing the data analytics. It also makes sure interoperability exists between the multi-vendor products for cross-layer data optimization.

#### 2.3.4. System Architecture View

The system architecture view contains multiple node views for the scalable fog deployments. It addresses the issues of technical teams, manufacturers, and system architects. The Performance and Scale vertical layer and some horizontal layers covered under the red border line in Figure 6 highlight the system architecture view of OpenFog RA.

*Hardware Platform Infrastructure:* It highlights the fog platform requirements for ensuring the safety of people and hardware from any harm, protection of the system from the environment, and mechanical support of the overall hardware infrastructure. The deployed system should also follow compliance and regulation standards.*Hardware Virtualization and Containers:* The hardware virtualization enables multiple entities to share the same physical machine and ensure system security by limiting specific system components from virtual machines (VMs). The use of containers decreases the overheads and provides lightweight mechanisms in the fog computing environment.

#### 2.3.5. Software Architecture View

It contains the architecture view of software running on a platform. The platform is formed with the combination of node views for addressing specific deployment scenarios. The fog node software is further separated into three layers, as shown in light blue colored layers in Figure 6.

*Application Services:* This layer provides the services with the help of other layers to accomplish the use case and specific requirements.*Application Support:* This infrastructure software part does not perform any new services but supports other applications in carrying out specific tasks.*Node Management and Software Backplane:* It performs node management and enables communication between nodes.

### 2.4. Comparison of RAMI 4.0, IIRA, and OpenFog Reference Architectures

The reference architectures given by different organizations have different approaches for the development and implementation of Industrial IoT. While RAMI 4.0 is mainly about the manufacturing process from the Production level to the Enterprise level, IIRA is about the industrial process with an established communication between deployed systems. The Platform Industrie 4.0 and IIC are currently collaborating to provide a common reference architecture by mapping the RAMI 4.0 and IIRA together [41]. While the RAMI 4.0 establishes the communication between the hardware and software by using a gateway, the IIRA provides the Edge Tier for the computation and storage of data. The OpenFog RA is about high data generation and processing use cases in industrial applications. OpenFog is designed to be implemented in any vertical integration application in the industry [12]. The selection of a particular reference architecture depends on the requirements of the deploying system. Table 2 shows the comparison of IIoT reference architectures.

## 3. Key IIoT Requirements, Emerging Technologies, and Literature Review of IIoT Architectures

As the IIoT is itself emerging due to the integration of Information Technology (IT) and Operational Technology (OT) [43], the problems due to its arising issues have to be addressed with the help of emerging technologies as well. The RAs such as RAMI 4.0, IIRA, and OpenFog provide the basic layout guidelines for the IIoT applications; however, due to the problems arising from heterogeneous technologies and diverse industrial usage, it is difficult to address the arising challenges just by following the reference architectures. In this regard, we have reviewed the IIoT architectural research papers to highlight the main IIoT requirements addressed in the current literature. The literature is solving the challenges for the full integration of Industrial IoT by using the various emerging technologies such as edge/fog computing, Software-Defined Networks (SDN), blockchain, 5G, Machine Learning, and WSN, along with the support of reference architectures, cloud services, protocols, and standards. Before discussing the literature review of IIoT architectures in detail, we highlight the key IIoT requirements and the emerging technologies used to address these challenges in the IIoT architectures.

### 3.1. Key IIoT Requirements

As the overlapping of Industrial IoT, Industry 4.0, and IoT is improving the production efficiency in industries, some challenges need to be addressed [15]. According to the RAMI 4.0 model, physical and virtual components of a deployed system can directly communicate with each other irrespective of the network hierarchy [10]; however, the system will require the interoperability ability for the system elements to communicate with each other. Due to the exponential growth of heterogeneous technologies, IIoT is facing many challenges in interoperability, latency, security, privacy, and scalability [27]. According to IIC in [44] and the authors of [45], security, privacy, and reliability are among the system characteristics and challenges in Industrial IoT systems. ITU has also defined latency, scalability, security, and privacy as the key requirements in IIoT networks [46]. Anitha et al. in [5] emphasized that IIoT requires high scalability compared to the IoT and highlighted the need for low network latency, interoperability, reliability, security, and privacy in IIoT in their research. Based on the challenges and information available in the literature, we have grouped the following key Industrial IoT requirements, which are critical for its full-stack development and integration in real-time.

*Interoperability:* Interoperability is the ability to share meaningful information between the two or more communication components [47]. In [48], the authors have highlighted the need for interoperability to guarantee the complete integration of industry 4.0 technology. Due to the increasing use of heterogeneous devices, technologies, and standards in industry 4.0, interoperability has become a major challenge for the industrial ecosystem [49]. The authors in [50] have further emphasized addressing the interoperability in IIoT for enabling the communication between the systems from individual vendors.*Scalability:* Scalability is the ability of a system to handle the increasing amount of work due to the growth of components throughout the system operation without affecting its performance [51,52]. According to [53], it is necessary to address the scalability solutions to deal with the exponential growth of devices and data generating in IIoT. The authors in [54] further highlight the need of scalability in IIoT and the main issues that affect it, for example, the diversity of networks, heterogeneity of devices, and massive data generated in IIoT systems.*Security:* As the IIoT is developing with the integration of both Information Technology (IT) and Operational Technology (OT), the current development of IIoT systems brings new security challenges, which cannot be addressed by using the traditional IoT security mechanisms [53]. According to Jamai et al. in [55], most of the security attacks in IIoT are focused on industrial devices, control systems, and networks. The authors in [56] have further classified the attacks on IIoT connectivity protocols into five threads: DoS/DDoS attacks, Information Gathering, Man in the Middle attacks, Injection attacks, and Malware Attacks.*Privacy:* “Privacy is the right of an individual or group to control or influence what information related to them may be collected, processed, and stored and by whom, and to whom that information may be disclosed” [44]. With the growing number of heterogeneous devices, it is essential to focus on data privacy issues in IoT and IIoT [57,58]. Different remedial frameworks are present in the literature to address the security and privacy issues in IIoT. According to [59], fog computing addresses the security and privacy issues in the IIoT, while the authors in [60] highlight the features of blockchain for solving the security and privacy issues.*Reliability:* Reliability in IIoT is the performance indicator that highlights the system working ability as per the design and for the specified time duration in industrial environment [44,61]. ITU has defined reliability as the essential ability for IIoT networks to avoid the risks and production interruptions [46]. The authors in [62] have presented the detailed literature review on the challenges of reliability in Devices, Networks, Applications, and Systems in IoT applications. A system is reliable if all of its components satisfy the reliability conditions.*Low latency:* According to ITU, network latency is the duration of time an information packet takes to reach the destination from the source [63]. According to the authors in [64,65], IIoT services are suffering critically from latency issues due to the generation of a huge volume of data. To address the latency issues, researchers are proposing solutions using different technologies such as 5G [66] and edge/fog computing [67].

### 3.2. Emerging Technologies used in Industrial IoT Architectures

In the literature review, we have found some similarities between the Industrial IoT architectures. The architectural solutions are developed by using some emerging technologies for the flexible integration and better performance of IIoT systems. We have grouped the widely used emerging technologies in the literature and focused on evaluating the scope of each technology in addressing the main IIoT requirements in those architectures. The following are some of the emerging technologies we have observed in developing the IIoT layered architectures:*Edge/Fog Computing:* Fog computing brings the cloud services closer to the ground mobile devices to offload the processing burden, improve the Quality of Service (QoS) of a system, and save resources [68]. Based on the information given by the National Institute of Standards and Technology (NIST), the fog computing should have the main characteristics of supporting the geographical distribution, low latency, interoperability features, scalability, and real-time interactions rather than batch processing [69]. The size of fog computing is smaller than the traditional cloud computing; however, the number of nodes can be combined to make it a large fog system [70].With the generation of an exponential volume of data from the sensors, it is difficult to process information locally due to the limitations of hardware devices. Edge computing provides the features to process the data at the edge device and reduce the required network resources for cloud computing by only sending the required data to the cloud for further processing [71]. Edge computing provides the data storage service at the edge, performs the tasks in the absence of cloud computing, and improve the network latency [72].*Software-Defined Networking (SDN):*The Software-Defined Networking (SDN) dynamically manages the distributed network segments to provide optimization and agility in a network with the help of programmable controlling units [73]. According to IBM, SDN provides dynamic load-balancing in network traffic and vendor-independent support with the ease of central programmability and configuration features [74]. SDN is based on three-layer architecture: Infrastructure layer (Data Plane), Control layer (Control Plane), and Application layer [75].*Blockchain:* Blockchain technology is based on decentralized and distributed nodes where all the transactions are processed after validation from the participants. In the blockchain, there is no third-party organization to control the transactions process, and the transactions from each participant are locally available to all the participants in the distributed ledger network forming data transparency [76]. According to [77], transparency and trust, decentralized networking, immutable data, and security are the main advantages of blockchain technology.*Machine Learning (ML):* Machine Learning (ML) is a subset of Artificial Intelligence (AI) that imitates intelligent human behavior based on accuracy with the help of data and algorithms [78,79]. ML has many applications, including prediction, semantic analysis, natural language processing, information retrieval, and computer vision [80]. According to research in [81], ML provides some necessary features in Industry 4.0, such as fault detection, predictive maintenance, security and threat detection, and human–machine interaction.*5G:* According to ITU, 5G is the evolution of previous mobile technologies (2G, 3G, and 4G) to deliver more speed for processing the high volume of data transfer with minimal latencies while also providing the large-scale connectivity for the exponential growth of devices and services [82]. As per the ITU’s recommendations for the International Mobile Telecommunications (IMT) for 2020, 5G technology has three main usage scenarios, 1) Enhanced Mobile Broadband (eMBB), 2) Massive Machine-type Communications (mMTC), and 3) Ultra-reliable and Low Latency Communications (URLLC) [83]. According to ETSI, 5G is facilitating new services in different domains, e.g., Industry 4.0, Education, Agriculture, and Publication Safety [84].*Wireless Sensor Networks (WSN):* According to the International Electrotechnical Commission (IEC), Wireless Sensor Networks (WSN) are the key IoT technology containing a large group of sensor nodes that detect the properties of physical phenomena such as temperature, humidity, light, pressure, etc., with the easy, reliable, and rapid deployment of systems [85]. In WSN, the nodes interact to form a cluster to utilize resources, providing network scalability and transmitting the collected data until it has arrived at the base station [86].

### 3.3. Current Research on IIoT Architectures

In IIoT architectures, we found some common topics in terms of challenges and emerging technologies. The layered architectures address key requirements with the help of emerging technologies for the end-to-end development of IIoT systems. We present a state-of-the-art review on how the layered architectures address these requirements by grouping them based on edge/fog computing, blockchain, SDN, 5G, Machine Learning, and Wireless Sensor Networks (WSN) technologies. Moreover, in references covering more than one technology, we grouped it with the more emphasized technology according to the paper, and papers not using any highlighted technology are included in Section 3.3.7.

#### 3.3.1. Edge/Fog Computing

To improve the lack of predictive maintenance in IIoT applications, reference [87] proposes the smart-machine maintenance model based on edge and cloud computing that addresses network scalability, security, and low latency. The proposed system also addresses the need for a high volume of data for suitable and trained algorithms with the help of a three-tier IIRA model. The fleet of machines containing several nodes sends the data to the edge device that performs data analytics using ML algorithms to send the diagnostic information to the platform tier for user monitoring. The proposed architecture uses machine learning for predictive maintenance but not for addressing the key challenges.

Due to the massive and diverse data generation in manufacturing industries, cloud services are unable to take care of large-scale data processing. Furthermore, the delay-sensitive information is vulnerable due to the semi-secure nature of cloud services. In this regard, Sengupta et al. have proposed an Industrial IoT architecture based on fog computing technology. The proposed solution is based on four layers perception layer, fog nodes layer, cloud layer, and application layer. To process the data and reduce the workload from cloud computing, the authors have included the fog nodes layer with semi-secure cloud computing features where a node can be a PC, a Raspberry Pi device, or a virtual operating system (OS). The authors have carried out the experiments in simulations as well as by developing a hardware testbed; however, the proposed solution does not address the reliability as per the harsh industrial environments and interoperability for accommodating the heterogeneous field devices [88].

In [89], the authors have addressed the system reliability shortcoming by presenting a fault-tolerant IIoT architecture using an edge gateway that also provides the low-latency, scalability, and security based on the industrial requirements. In the practical example, the authors have developed a system for machine operative status detection using the raspberry pi as an edge device that stores the information in the local database. The edge device uses this data with algorithms to predict the machine status and display the monitoring parameters such as current, power consumption, and vibration. With edge computing, the proposed system avoids the congestion of bandwidth, unnecessary network lags during the data transfer, and securing the information by bringing it closer to the edge.

The authors in [67] present a conceptual architecture intending to integrate versatile fieldbuses and solve interoperability issues. The proposed model ensures data security by bringing the data processing closer to the edge/fog nodes. Furthermore, the ability of distributed edge/fog nodes in different domains provides high network scalability. The proposed model also addresses the reliability and low latency of the communication process and is based on four layers—Sensing layer, Data Provider layer, Fog/Edge Computing layer, and Application/Services layer. The Sensing layer contains peripherals and devices connected to specific fieldbuses such as Modbus and Ethernet. The Data Provider layer stores the bidirectional data from fieldbuses and upper layers in the buffer memory, while the Fog/Edge computing layer performs the data processing. The Applications/Services layer provides developed applications for remote monitoring and controlling. The authors have emphasized interoperability for M2M communication between the network elements; however, this conceptual model does not address the data privacy concerns.

The function of distributed automation systems in industries with heterogeneous technologies, protocols, and devices from different vendors is the future of industrial processes; however, the high number of connected devices in current systems are having privacy and interoperability problems with exchanging the information efficiently. In this regard, Dobaj et al. have proposed a state-of-the-art lightweight, flexible, and secure Industrial IoT theoretical architecture with the continuous system integration and development (CI/CD) process under the containerized environment. The use of distributed edge/fog nodes allows minimum latency and network scalability. Furthermore, the proposed microservices-based architecture ensures network reliability with the support of fault-tolerant network protocols such as OPC-UA, and DDS. The data privacy is ensured by keeping the data at the respective microservice unit and can only be accessed by using its API [90]. The authors have addressed all the IIoT challenges we have highlighted in our paper; however, they have proposed the architecture based on a theoretical approach, not by performing hardware or simulation-based experiments.

#### 3.3.2. Software-Defined Networking (SDN)

According to ref. [91], the performance of the IIoT depends on the deployed systems and the set of communication protocols. Furthermore, the efficiency and reliability of the existing IIoT architectural solutions are compromised due to the lack of testing and usage of new protocols. This problem has resulted in the integration of SDN technology with the IIoT architectures. Moreover, as the IIoT devices are generating high data, the transmission of information is facing delays and causing computation offloading issues. In this regard, Chandramohan et al. in [92] have used Software-Defined Networking (SDN) emerging technology in their proposed architectural solution for efficient Quality of Service (QoS)-based communication. SDN provides the priority-based transmission control with low processing time performed at the edge device. The edge allows the features of adaptive computing and network scalability. In the proposed architecture, the physical layer contains various nodes with a single cluster head as the main device, which interacts with the control layer. The centralized SDN controller manages the network flow routing and provides access to the user application. The proposed model is simulated in MATLAB, and the results showed the advantages of network reliability, higher throughput, and lower latencies.

The OPC-UA network protocol in client–server communication provides the Machine-to-Machine (M2M) information exchangeability; however, the traditional devices do not support this protocol. To solve the interoperability issues and provide reliable and low latency-based communication, the authors in [93] have proposed OPC-UA gateways and Time Sensitive Software-Defined Networking (TSSDN)-based Industrial IoT architecture. The network elements send the information to the OPC-UA-based edge gateway that handles the heterogeneous data and enables communication between the vendor-specific devices. The TSSDN switch enables reliable and low latency-based communication by controlling the network resources. The proposed architecture showed efficient results of information exchange between the network components; however, the authors have not addressed security, privacy, and scalability requirements in the given architecture.

Bedhief et al. in [94] have proposed a software-based architecture for IIoT based on SDN and edge/fog computing technology. While SDN provides flexibility and scalability, fog/edge computing enables low latency and interoperability. The central programmability approach of SDN in the proposed solution allows the flexibility to use the heterogeneous network technologies, which can be deployed and changed independently. However, the authors have not addressed the security and privacy features in the proposed architecture.

The security and privacy shortcoming is improved by Friha et al. in [95] by using SDN technology with Blockchain’s Hyperledger Sawtooth and fog computing. The proposed robust framework contains four layers specifically for the secure Agricultural IoT. (1) The Agricultural layer contains the peripherals for sensing and controlling, (2) the Fog layer contains various nodes that provide the storage, data processing, and computations in a containerized docker environment near the end devices, (3) the SDN Controller Network layer contains the central controller, and all networks act as a single Network Operation System (NOS), and (4) Blockchain Network layer, which validates all the information and enforces the transactions in the system by establishing trust using Distributed Ledger Technology (DLT). However, the proposed architecture does not address interoperability.

The future of industries is to be accompanied by the constellation of thousands of sensors and devices. Without the interoperability between heterogeneous devices, the deployed systems will be handled by various vendor-specific solutions that will create the problems of not utilizing the performance of system elements collectively. In this regard, the authors in [96] have proposed an open-source Software-Defined Networking (SDN)-based IIoT architecture with the OpenDaylight (ODL) SDN controller. The proposed architecture contains three layers: Data Plane, Control Plane, and Application Plane. The data plane layer is composed of switches, routers, and other network devices forming the SDN and WSN network, and it handles the traffic flow based on Quality of Service (QoS) and takes care of the data routing. The Control Plane sends the information to the Application Plane that manages the SDN operations and provides the cloud services and controlling features. While WSN provides scalability, ODL further ensures the fault-tolerance and scalable network with the central control of a group of controllers. The given IIoT architecture also provides network reliability and fault tolerance by monitoring and providing the redundant ODL controller features.

#### 3.3.3. Blockchain

In [97], the researchers suggest blockchain technology to make the processing chain in Industrial IoT secure, traceable and transparent. Teslya et al. have proposed a conceptual blockchain-based model for security, trust in the network, and reliability; however, the proposed model does not address the interoperability and has its drawbacks of the durability of information in Semantic Information Broker (SIB); and non-matching of data between the different participants [98].

The authors in [99] have addressed security and privacy challenges in their theoretical blockchain-based IIoT architecture. The proposed model ensures the addressed shortcomings by establishing trust between the components. The message transactions in this solution are secured by using the gossip protocol-based private/public key exchange between the communication nodes.

In industries, sensors lack the capabilities to process, compute, and detect security vulnerabilities. Furthermore, the current solutions lack authentication, integrity, and identification ability. In this regard, the authors in [100] have presented a practical distributed ledger-based authentication framework. The proposed framework utilizes the combination of Secure Multi-Party Computation (SMPC) and Distributed Ledger Technology (DLT) to detect attacks and malicious sensors in Industrial IoT. The distributed ledger technology solves the aforementioned issues in a decentralized way, establishing the trustworthiness of sensors by implementing a consensus mechanism at each node; however, the theoretical and practical models presented in [99,100] have a shortcoming in terms of handling the large-scale devices, which will create scalability problems.

Lin et al. in [101] have addressed this shortcoming by combining the Oracle software features with blockchain technology. Blockchain technology, in the literature, provides trust and ensures security; however, the current blockchain-based decentralized architectures cannot obtain complex real-time and isolated data with low processing time. In this regard, the authors have used Federated Learning (FL) with Oracle and blockchain to propose IIoT digital twin architecture that provides a low processing time and high network traffic stability. The oracle-based fast computing mechanism allows the exchange of trusted data between the physical and digital machines in a decentralized network.

Ghajar et al. in [102] have further addressed the interoperability and trust challenges along with security and privacy features by proposing Schloss, a blockchain-based IIoT architecture. The proposed architecture authenticates the network nodes based on the application-level authentication process in the distributed blockchain management system. The model ensures the nodes’ privacy, whereas the authority of each node is decided based on its behavior. The architecture contains a feature to decrease the node power based on the Proof of Work (PoW) between the nodes. The proposed model ensures network security and establishes trust between business partners. The devices connected to the network are dynamically identified and controlled by using the multi-signature intelligent contract mechanism while maintaining data privacy.

The integration of distributed ledger technology (DLT) feature provides data security and privacy; however, the current IIoT architectures based on blockchain are subject to scalability, latency, and computational resource issues. In this regard, the authors in [103] have proposed a lightweight hash function-based IIoT architecture to improve the latency and scalability issues for devices with low power and processing specifications. The network comprises a group of cell nodes responsible for validation and ledger management. The results show this architecture can improve the scalability and latency compared to other blockchain-based models; however, the proposed solution does not address issues compared to architectures based on other emergent technologies.

In [104], the authors have addressed the scalability and latency along with security and privacy challenges by proposing fog computing and blockchain-based security architecture for IIoT-enabled Cloud Manufacturing (CM). The authors have focused on addressing three main things that are lacking in the security of CM in the current literature, (1) trust between the network nodes to ensure the authenticity by using the blockchain-enabled Elliptic Curve Qu Vanstone (ECQV) certificates, (2) privacy of CM data over the internet, and (3) scalability requirements of security services to deal with future expansions.

The IIoT architectures with centralized controlling mechanisms are being targeted by various security attacks due to the use of different network technologies. In this regard, the authors in [105] have proposed security, privacy, and trust ensuring architecture with the help of lightweight and decentralized ledger technology. The Proof of Authentication (PoU) mechanism manages the trusted and secured communication between the nodes. The proposed decentralized solution is lightweight, scalable, and efficient for resource-constrained IIoT devices.

The use of heterogeneous technologies is resulting in privacy and security issues between the network components, and that is also causing a lack of trust among the participants. To address these challenges together with scalability, low latency, and network reliability, Ceccarelli et al. in [106] propose an Industrial IoT architecture, specifically for real-time railway systems, by combining blockchain, fog computing, and SDN emerging technologies. The computing nodes in the proposed FUSION model are reconfigurable to act as Fog/Edge, SDN, or End Devices based on the system requirement. The blockchain ensures the information exchange between the decentralized network components in a secured and trusted environment. The SDN technology in the given architecture allows the network resources management and reconfiguration of system operations. Furthermore, edge/fog computing ensures low network latency and provides information processing and storage closer to the devices. While the blockchain enables secure and privacy-preserved communication, the decentralized control of system architecture with SDN ensures network scalability.

#### 3.3.4. Machine Learning (ML)

The Industrial Control Systems (ICS) form the basis of intelligent industrial sectors; however, due to the integration of operational technology (OT) and information technology (IT), these industrial sectors are subject to security threats, which are necessary to address. To address these issues and provide a futuristic unified solution, the authors in [107] have proposed an IIoT reference architecture based on five-zone layers, from the Experimental layer to the Management layer. The layers contain ICS elements such as Supervisory Control And Data Acquisition (SCADA) and Programmable Logic Controller (PLC) to interact with field devices using Remote Terminal Unit (RTU), Modbus, and Open Platform Communications United Architecture (OPC UA) protocols. Different cyber security datasets are also reviewed and presented for using them with machine learning algorithms for network security. The proposed model also contains a theoretical case study of solving the interoperability issues within heterogeneous systems with the help of a VPN router. The authors have presented this conceptual architecture without highlighting any experimental work while designing a testbed system for a cyber security group under the national infrastructure project.

The Android operating system (OS) has recently been facing a lot of malware attacks due to its integration with heterogeneous IIoT devices. There are various ML-based solutions to provide security; however, the models in the literature rarely address data privacy. Since the algorithms are trained in a centralized way where all the network nodes have to share their data, it is causing privacy issues. In this regard, Taheri et al. have proposed a Federated Learning (FL)-based decentralized privacy protection architecture for Industrial IoT. The network nodes do not have to share private information with the FL approach and train the algorithms locally using the global training model. The authors have also addressed the vulnerabilities of traditional FL-based solutions in the current literature that are susceptible to security attacks from the participants’ side while they are in the learning phase. To address the shortcomings of FL in the literature and evaluate the efficiency of the proposed architecture, the authors have proposed an architecture in two parts; the first part contains the poisoning attacks based on the Generative Adversarial Networks (GAN) and Federated GAN. For the counter-measure solution, the authors have utilized Byzantine Median (BM) and Byzantine Krum (BK) to detect these malware attacks and to ensure network reliability at the server-side. The proposed architecture provides 8% more accuracy than the existing architectural solutions [108].

As the IIoT is growing due to high-scale data sensing, processing, and storage, many adversarial attacks are breaking the security barriers to access the user data, steal it, and inject different malware and other malicious codes. Some of the increasing attacks are DoS, DDoS, Advanced Persistent Threat (APT), and modern botnets. To solve these issues, the authors in [109] have proposed a Convolutional Neural Networks (CNN)-based botnet and malware detection architecture to ensure security and privacy while also addressing the interoperability and scalability at the network layer. The proposed architecture uses the hybrid long short-term memory and CNN-based DL approach by utilizing the publicly available datasets. It provides efficient results in terms of accuracy and speed.

The integration of IIoT with Industrial Control Systems (ICS) is bringing new changes in manufacturing processes with preventive maintenance; however, the IIoT systems are under constant security threats from all the architectural layers. In this regard, the authors in [110] have proposed a machine learning and blockchain-based IIoT architecture for intelligent manufacturing systems. In the proposed solution, the security threats at each architectural layer are highlighted together with their possible remedial solutions. The experiments showed that blockchain technology and machine learning algorithms reduce the number of attacks compared to standalone ML algorithms.

#### 3.3.5. 5G Technology

Ludwig et al. have proposed a 5G architecture based on the 5G use cases in various industries such as Smart Production, Condition Monitoring, Distributed Sensing, and Automated Guided Driving. The proposed architecture consists of different edge devices connected to the public and private base stations via eMBB, uRLLC, and mMTC wireless mechanisms of 5G. The authors have also included the Software-Defined Networking (SDN) in their architecture for reliable communication using effective management of network resources [111].

The authors in [112] have further addressed network scalability in their proposed solution. Due to the high number of IIoT devices, the existing architectures are not providing low latency and reliable communication with high scalability. This issue has resulted in the creation of Mobile Edge Computing (MEC); however, the MEC-based architectures present in the literature face a diverse nature of components and technologies, complex development of IIoT systems, lack of flexibility, and poor mobility. In this regard, the authors have proposed the MEC architecture by combining the docker container technology. The runtime instances of the docker images run independently, and the containers map the physical components with the virtual environment. While the 5G provides low latency and reliable communication, the docker containerization makes the mobility of the proposed architecture efficient and ensures high scalability.

The authors in [113] have addressed more key IIoT requirements in their proposed conceptual architecture by addressing the security and privacy features along with low latency, scalability, and reliable communication. The proposed framework architecture for smart manufacturing addresses the IIoT requirements based on its six architectural layers.

Wang et al. in [114] propose an experimental Quality of Service (QoS) and secure privacy preserved Industrial IoT architecture based on 5G technology and Federated Learning. The 5G brings reliability and low latency, while the FL further improves the latency and deals with load-balancing and privacy leakage issues. The minimum possible routing paths are selected in the model to attain the minimum latencies. As in [113], this proposed solution addresses many requirements; however, it does not address the interoperability features for the reusability of data and machine-to-machine communication in IIoT systems.

According to Jiang et al. in [115], the communication among the network elements is not secured until the trustworthiness of all partners is ensured. In this regard, the authors in [116] have combined a trust and authentication method in their proposed 5G technology-based architecture for the network components to cope with security and privacy issues due to the exponential growth of data. The proposed solution uses an Advanced Encryption Standard (AES)-based encryption method to ensure the secure data transfer between the participants. Furthermore, the Dempster Shafer Theory (DST) method in the architecture allows the reliability and trustworthiness of the collected data from sensors. While 5G technology provides a high bandwidth for low latency, the network scalability is achieved by using the gateway with the help of a cloud server.

In [66], the authors have proposed a 5G-enabled IIoT architecture named Smart Networks for Industry (SN4I) to address the increasing use of Industry 4.0 in industrial manufacturing. The proposed architecture addresses the interoperability and heterogeneity issues such as lack of dynamicity due to the static utilization of components for a fixed solution. By enabling network interoperability, this architecture ensures the reusability of resources. It secures Wireless Sensor Networks (WSN) by blocking unauthorized access within the network using the Hydra Server access control protocol mechanism. The SDN and NFV technologies in the proposed solution ensure the interoperability and scalability of the system. Moreover, Wireless Sensor Networks (WSN) technology is also used to further improve network scalability.

#### 3.3.6. Wireless Sensor Networks (WSN)

In [117], the authors have proposed a general-purpose two-tier wireless architecture for the reliablity and ease of implementation efforts of Industrial IoT. The upper tier in the proposed model is responsible for the information exchange between the network nodes based on wireless and wired communication. The QoS configuration of the switch allows the control of communication and bandwidth quality, whereas the communication is possible with the help of TCP/IP/UDP protocols. The architecture is suitable for the MODBUS and OPC-UA-based communication between the machines. The lower tier contains the Head Devices (HD), which interact with the controllers such as PLCs. Low power and reliable communication are achieved by employing the 6TiSCH-based frequency hopping technique with the ubiquitous connectivity based on IPv6 with Wireless Sensor Networks. The authors have tested the proposed architecture by using Raspberry Pi as the Head Device (HD) connected to the remote I/O terminals using the M2M protocols.

Due to the centralized handling and processing of information by a single centralized controller in Industrial Wireless Sensor Networks (IWSN), the IIoT systems experience security, privacy, and latency issues. In this regard, Benomar et al. have proposed a decentralized IWSN and fog computing-based architecture. Different devices and sensors from the factory floor level send the collected information to their respective fog nodes called motes. The fog nodes use Singular Value Decomposition (SVD) schemes for low latency and higher throughput. Moreover, the transmission between network components is secured with the help of a lightweight ciphering technique. The results from this implemented solution showed an efficient packet delivery ratio and latency [118].

The ubiquitous connectivity of large-scale wireless sensor networks (WSN) requires high manageability from the sensors to the application level in IIoT. In this regard, the authors in [119] have proposed WSN and software-defined networking (SDN)-based IIoT architecture comprising the number of interconnected wireless field devices (FD) to ensure scalability using the WirelessHART protocol. The heterogenous FDs are connected to the central gateway and achieve interoperability using the Constrained Application Protocol (CoAP). The OpenFlow SDN controller ensures the QoS of information exchange between the gateway and cloud securely and reliably using the WebSocket protocol. Based on the experiments, SDN QoS-based information exchange shows better latency over non-SDN communication between the gateway and cloud.

#### 3.3.7. Miscellaneous Architectures

In IIoT systems, the use of heterogeneous communication protocols is causing the lack of communication between the cloud and the devices at the field level. To address this issue, the authors in [120] have proposed an IIoT gateway architecture in which the smart gateway carries out the protocol conversion processes. The gateway exchanges information from factory floor objects using different protocols such as Modbus, MQTT, S7, OPC UA, and BACnet and enables the interaction between cloud and floor devices in a unified way by using the MQTT protocol. The protocol conversion process is reliable with the help of asynchronous processing mechanisms for task scheduling and real-time operation. The communication is secured with the help of encryption mechanisms in the Network, Transport, and Application layers. Based on the experimental results, the solution provides a reliable and secure transfer of information from low-level devices to the cloud.

The traditional IIoT architectures are vulnerable to trust issues due to the large number of connected objects where a single infiltration can lead to the failure of a complete security system. Moreover, distributed trust management is also ineffective under industrial scenarios. In this regard, the authors in [121] have proposed the trust management-based IIoT architecture with the concept of industrial relationships. In the proposed model, the industrial network is composed of clusters called communities that exchange valuable information by electing trusted leaders from each group to ensure privacy. The leader of each community calculates the trustworthiness of connected devices based on direct/indirect honesty and cooperation features. Based on the simulations performed, the proposed architecture shows convenient results of trust management between the components and provides adaptiveness and resiliency features.

## 4. Observations and Discussion

### 4.1. Experimental vs. Conceptual Architectures

Aside from reference architectures, we have also reviewed the proposed IIoT architectures by extracting them with general keywords (“IIoT” or “Industrial IoT” or “Industrial Internet of Things” and “architectures”) from different literature databases. The research papers obtained were further shortlisted, where the researchers presented layered architectures with end-to-end features from the industrial floor level to the enterprise level. The final list of research papers reviewed here is from 2015 to 2022, which we have divided into two categories—experimental and conceptual architectures. In experimental architectures, the authors have performed real-time experiments on their proposed models either by testing and evaluating the hardware-based prototypes or by testing the simulations in the virtual environment. The conceptual architectures are based on theoretical knowledge without performing any experiments. Figure 7 shows the research trend by presenting the architectures from 2015 to 2022. The reference architectures have laid the foundation of proposed architectures in the literature, and the focus on providing experiment-based architectures is increasing over time.

### 4.2. Comparison of Proposed Architectures in the Literature

Researchers in the literature have proposed various architectures to address the main IIoT challenges and requirements. Table 3 highlights the work of each architecture in the literature reviewed by addressing the features of scalability, interoperability, security, privacy, reliability, and low latency for Industrial IoT. Based on the literature reviewed in this paper, there is a research gap in addressing all these requirements collectively. Although the IIoT architecture in [90] addresses all the features, the authors have presented this architecture based on the theoretical approach, not the practical.

Figure 8 shows the focus of current IIoT architectures on addressing the Industrial IoT requirements in the order of security, low latency, scalability, reliability, privacy, and interoperability. As Industry 4.0 is currently in its initial phase of development with the integration of Industrial IoT, the current literature needs to focus on interoperability for the efficient utilization of resources through machine-to-machine (M2M) communication.

### 4.3. Relation of Emerging Technologies to Key Requirements

Based on the literature reviewed in this paper, Table 4 shows the use of emerging technologies to address the main IIoT requirements in the literature. While [120,121] have proposed the architectures without using any of the mentioned emergent technologies, other papers have utilized one or more than one technology along with standards and protocols in their proposed architectures.

Apart from the relation between key IIoT requirements and emerging technologies, we also highlight the trend of these technologies in IIoT architectures. Figure 9 shows the use of emerging technologies in presenting architectural solutions. The current literature is focused massively on utilizing the processing and storage characteristics of edge/fog computing to provide IIoT architectures. The decentralized privacy-preserving and trust-establishing features of blockchain are also a hot topic in IIoT, followed by the agile controlling features of SDN and enhanced network speed of 5G. While WSN is addressing the high number of applications in IoT, more IIoT architectures prefer wired-based communication over the WSN technology, except for a particular use case or mobility requirements. Furthermore, the literature is least focused on presenting the architectures based on Machine Learning. Researchers are using preventive maintenance and smart algorithms of machine learning to address many specific solutions in Industrial IoT; however, the literature has yet to utilize the full potential of ML in addressing the challenges and forming end-to-end architectures.

The scope and characteristics of each emerging technology are unique in terms of addressing the challenges in IIoT architectures in literature. Figure 10 highlights the scope of each emerging technology in IIoT architectures. Researchers are using edge and fog computing to solve the main IIoT requirements; however, the current literature has not utilized this technology to address all the challenges collectively in an IIoT architecture. Blockchain technology highly addresses the security and privacy issues in IIoT architectures, while some literature also focuses on a few other challenges of scalability, reliability, and interoperability. We have also observed a unique characteristic of blockchain technology contrary to other emergent technologies, and this feature is called the trustworthiness of an architectural component. The trust between the network elements plays a salient role in achieving security and privacy-preserved communication. In the literature review, we have found that 11/34 papers address the trust feature in their proposed architectures. While 9/11 of these research papers address this issue using blockchain, the other two use the industrial relationship concept between the group of network clusters [121] and the data encryption technique [116].

The information from Table 4 highlights that research highly uses SDN in combination with other emergent technologies in the architectures. The central network controlling characteristics of SDN enables it to provide reliability scalability and low latency, while some literature has also used SDN for addressing interoperability and security issues. The adoption of 5G technology in IIoT architectures provides high-speed features with minimal latency compared to the other technologies. 5G also addresses the scalability and reliability challenges in IIoT architectures. The integration of Wireless Sensor Networks (WSN) technology in IIoT architectural solutions addresses three challenges—low latency, reliability, and scalability. WSN extensively addresses the scalability requirements as compared to other features. The use of machine learning (ML) in IIoT architectures preserves data privacy from unauthorized access and ensures network security.

## 5. Conclusions and Future Work

In this paper, we presented a state-of-the-art review on IIoT reference architectures from organizations and proposed architectures in the literature, the main IIoT requirements for end-to-end implementation, and the emerging technologies used in architectural solutions to address these requirements and challenges. Each reference architecture has specific characteristics of industrial use case applications, system topology, services, data processing, storage, and computation abilities. The selection of particular reference architecture depends on the required full-stack IIoT solution under specific industrial scenarios. We presented a systematic review transitioning from reference architectures to proposed architectures, providing the rationale for research from academia. We identified that the main IIoT issues addressed in various research papers are scalability, interoperability, security, privacy, reliability, and low latency. These are the main requirements that mainly affect the deployment of industrial IoT in real-time. We also identified the use of edge/fog computing, blockchain, SDN, 5G, Machine Learning, and WSN technologies in developing the architectural solutions and their unique characteristics in addressing the challenges. We also highlighted the literature focus on utilizing these technologies and addressing the challenges.

On the other hand, each IIoT architecture addresses at least two main requirements, either with a conceptual approach or with a simulations/hardware-based experimental approach. The authors in [90] have addressed all the mentioned requirements based on the theoretical model, not the practical solution. Meanwhile, the literature is trending towards presenting more experimental architectures over time. We have described the possible research directions that can contribute to the flexible deployments of IIoT systems. There is a need to provide a common IIoT architectural framework that addresses all the applications in IIoT under harsh industrial conditions and ensures secure and reliable integration from the factory floor up to the enterprise level. The future development of IIoT architectures will keep adding more research challenges driven by Augmented Reality (AR) and Digital Twins in the Industry 5.0 paradigm shift; and its integration with 6G-based connectivity features will result in more data storage, processing, and computational requirements for data analytics in intelligent industrial systems.

## Figures and Tables

**Figure 1 sensors-22-05836-f001:**
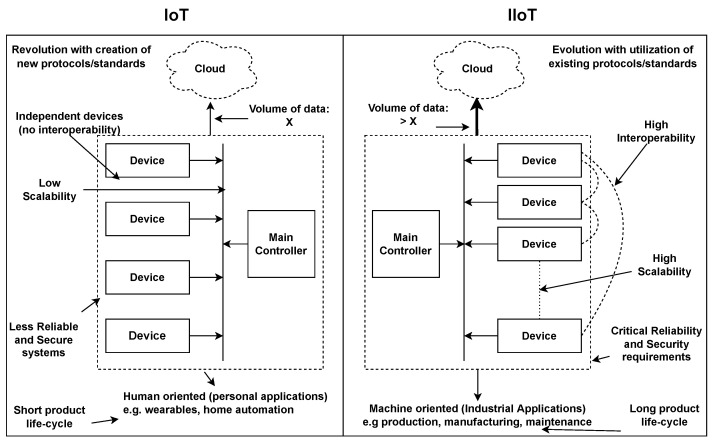
Main differences between IoT and IIoT [5,14,15].

**Figure 2 sensors-22-05836-f002:**
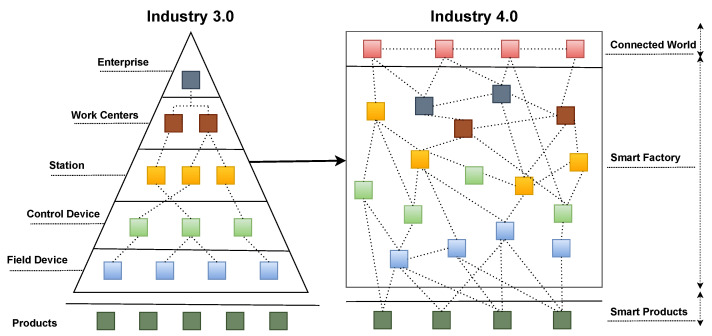
Industry 3.0 vs Industry 4.0 (adapted from [32]).

**Figure 3 sensors-22-05836-f003:**
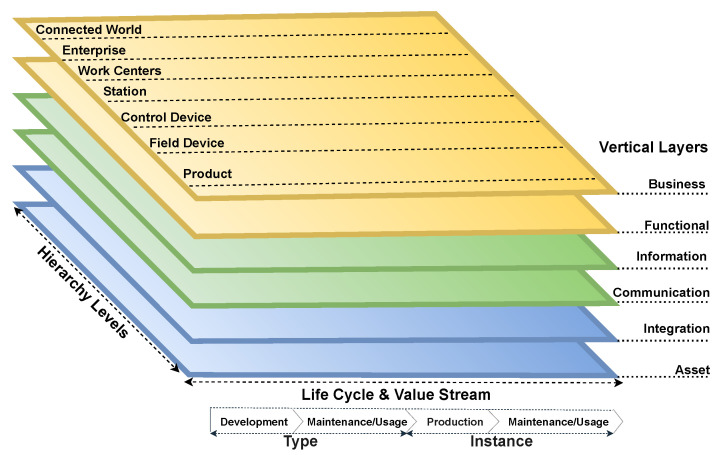
RAMI 4.0 architecture model (adapted from [10]).

**Figure 4 sensors-22-05836-f004:**
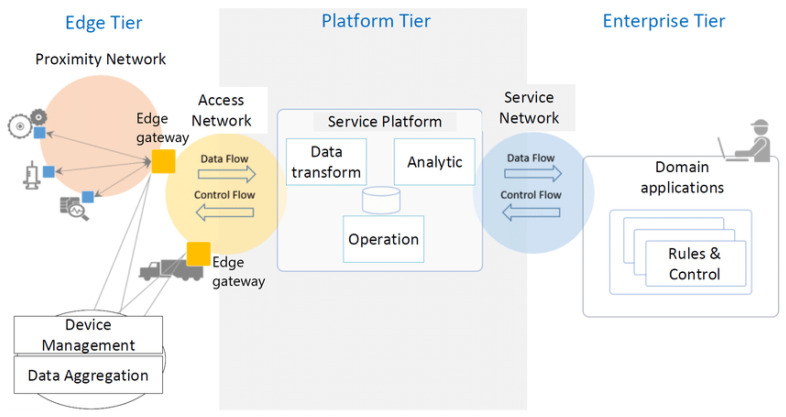
Three-Tier IIoT system architecture of IIRA (Reprinted with permission from [11]. Copyright 2019 Object Management Group).

**Figure 5 sensors-22-05836-f005:**
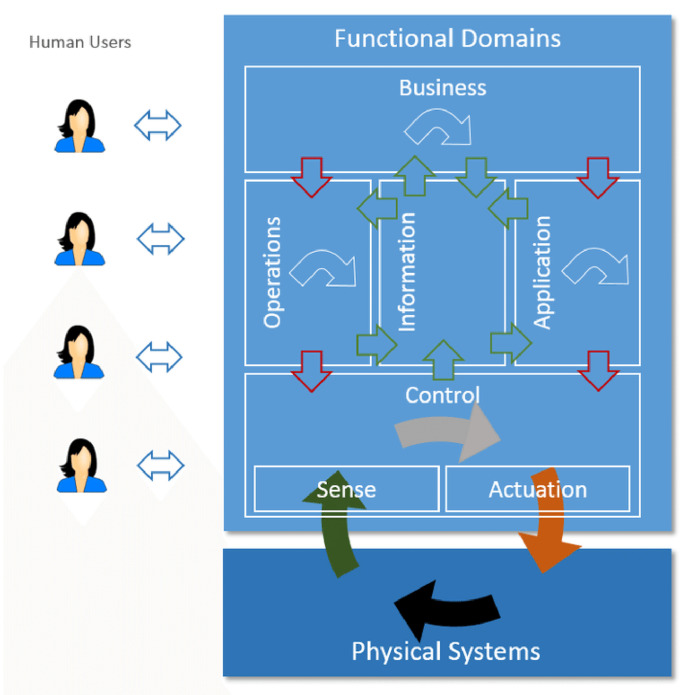
IIRA domains (Reprinted with permission from [11]. Copyright 2019 Object Management Group).

**Figure 6 sensors-22-05836-f006:**
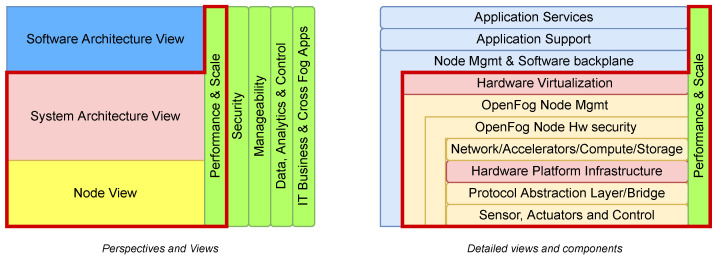
OpenFog reference architecture (adapted from [40]).

**Figure 7 sensors-22-05836-f007:**
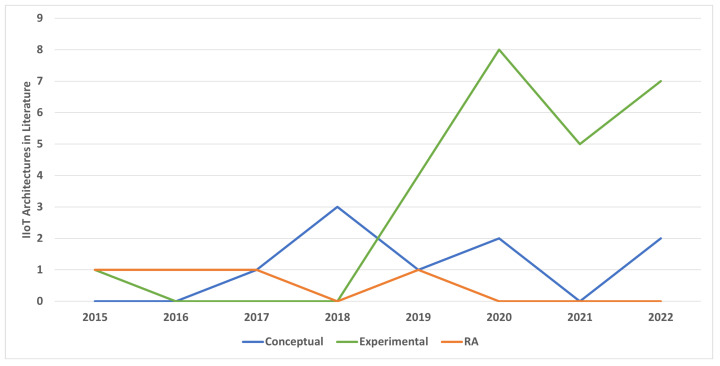
Conceptual and experimental architectures in the literature.

**Figure 8 sensors-22-05836-f008:**
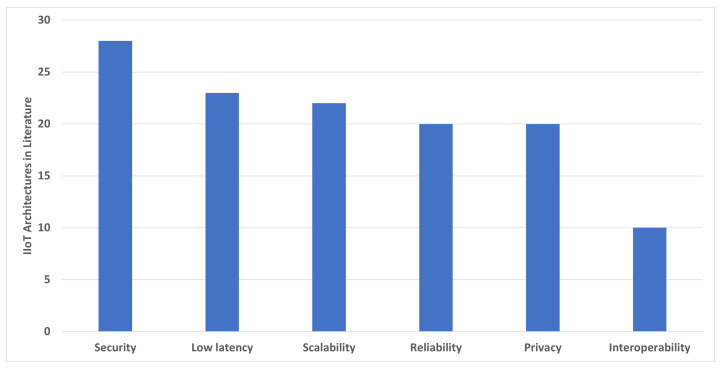
Literature focus on key IIoT requirements.

**Figure 9 sensors-22-05836-f009:**
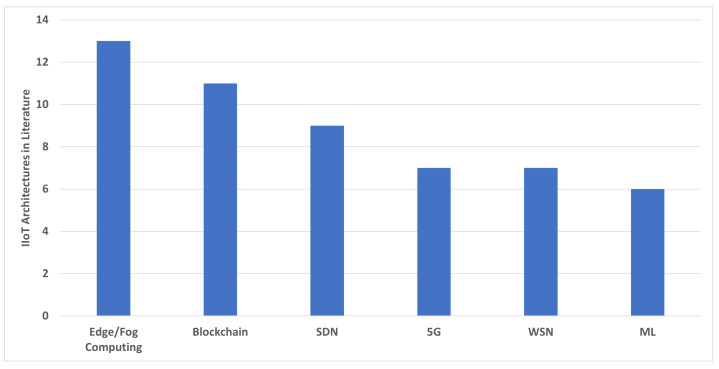
Focus on emerging technologies in the literature.

**Figure 10 sensors-22-05836-f010:**
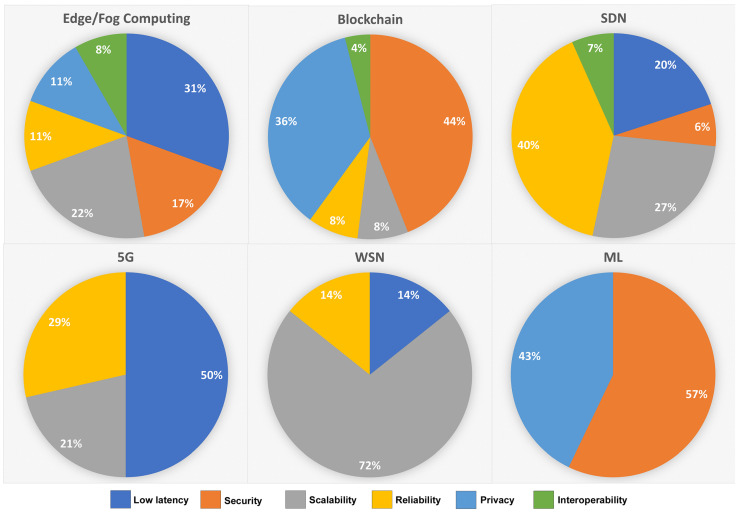
Relation between emerging technologies and key requirements in IIoT architectures.

**Table 1 sensors-22-05836-t001:** Related IIoT architectural review and survey papers and their main topics.

Refs.	Year	Arch. Type	Emerging Technologies	Challenges
[18]	2018	Reference		
[16]	2019	Reference		
[19]	2020	Reference		
[27]	2020	Reference and Proposed	Cloud/Fog Computing, ML, Blockchain	Scalability, Heterogeneity, Security
[23]	2020	Proposed	Blockchain	Security, Privacy, Scalability
[22]	2020	Proposed	Fog Computing, WSN	Low latency, Security
[17]	2021	Reference		
[20]	2021	Reference		Interoperability
[24]	2021	Proposed	Machine Learning, Edge Computing	Scalability, Low latency, Reliability, Security
[25]	2021	Proposed	SDN/NFV, 5G, WSN, Edge Computing	Scalability, Security, Privacy, Reliability, Low latency, Interoperability
[21]	2022	Proposed	Blockchain, 5G	Security, Privacy, Interoperability
[26]	2022	Proposed	5G, WSN, SDN, Blockchain, Edge Computing	Interoperability, Low latency, Security, Privacy, Scalability, Reliability

**Table 2 sensors-22-05836-t002:** Comparison of Industrial IoT reference architectures.

Category	RAMI 4.0	IIRA	OpenFog	Refs.
Organization	German Electrical and Electronic Manufacturers’ Association (ZVEI).	Industrial Internet Consortium (IIC).	OpenFog Architecture Workgroup.	[11,12,33]
Layers	Business, Functional, Information, Communication, Integration, and Asset.	Business, Usage, Function, and Implementation.	Included but not limited to Functional and Deployment viewpoints.	[12,41,42]
Hierarchy	Product, Field, Device, Control Device, Station, Work Centers, and Enterprise.	Not hierarchy-based.	Devices, Monitoring and Controlling, Operational Support, Business Support, Enterprise Systems.	[12,41]
Connectivity	Whitepaper	Framework	Framework	[12,41]
Difference in Industry Applications	Focused on manufacturing things smartly through Product Life-Cycle process.	Covers the manufacturing process but does not complete the product life cycle. Enables things to work smartly with the interaction of large deployed systems.	Focused on generic platform for applicability with any vertical market use case studies. e.g., agriculture, smart cities, transportation, etc.	[12,41]
Gateway, edge/fog	Analyze the data and connects the hardware and cloud at the gateway.	Computing, processing, and storage at edge.	Storage, Processing, Computing, Accelerators, and Network capabilities for vertical application at each fog hierarchy.	[12,39]

**Table 3 sensors-22-05836-t003:** Comparison of IIoT architectures in the literature.

Refs.	Arch. Type	Security	Low latency	Scalability	Reliability	Privacy	Interoperability
[99]	Conceptual	✔				✔	
[111]	Conceptual		✔		✔		
[107]	Conceptual	✔					✔
[98]	Conceptual	✔	✔		✔		
[102]	Conceptual	✔			✔	✔	✔
[67]	Conceptual	✔	✔	✔	✔		✔
[96]	Conceptual	✔	✔	✔	✔		✔
[113]	Conceptual	✔	✔	✔	✔	✔	
[90]	Conceptual	✔	✔	✔	✔	✔	✔
[100]	Experimental	✔				✔	
[108]	Experimental	✔				✔	
[120]	Experimental	✔			✔		
[110]	Experimental	✔				✔	
[103]	Experimental	✔				✔	
[92]	Experimental		✔	✔	✔		
[93]	Experimental		✔		✔		✔
[101]	Experimental	✔	✔	✔			
[112]	Experimental		✔	✔	✔		
[117]	Experimental		✔	✔	✔		
[121]	Experimental	✔			✔	✔	
[87]	Experimental	✔	✔	✔			
[105]	Experimental	✔		✔		✔	
[88]	Experimental	✔	✔	✔		✔	
[94]	Experimental		✔	✔	✔		✔
[104]	Experimental	✔	✔	✔		✔	
[109]	Experimental	✔		✔		✔	✔
[118]	Experimental	✔	✔	✔		✔	
[119]	Experimental	✔	✔	✔	✔		✔
[66]	Experimental	✔	✔	✔		✔	✔
[89]	Experimental	✔	✔	✔	✔	✔	
[95]	Experimental	✔	✔	✔	✔	✔	
[106]	Experimental	✔	✔	✔	✔	✔	
[114]	Experimental	✔	✔	✔	✔	✔	
[116]	Experimental	✔	✔	✔	✔	✔	

**Table 4 sensors-22-05836-t004:** Emerging technologies in the literature.

Refs.	Edge/Fog	Blockchain	SDN	5G	WSN	ML
[99]		✔				
[111]			✔	✔		
[107]						✔
[98]		✔				
[102]		✔				
[96]			✔		✔	
[67]	✔					
[113]	✔			✔		
[90]	✔					
[100]		✔				
[108]						✔
[120]						
[110]		✔				✔
[103]		✔				
[92]	✔		✔			
[93]			✔			
[101]		✔				✔
[112]	✔			✔		
[117]					✔	
[121]						
[87]	✔					✔
[105]		✔				
[88]	✔					
[94]	✔		✔			
[104]	✔	✔		✔		
[109]						✔
[118]	✔				✔	
[119]			✔		✔	
[66]			✔	✔	✔	
[89]	✔					
[95]	✔	✔	✔			
[106]	✔	✔	✔			
[114]				✔	✔	✔
[116]				✔		

## Data Availability

Not applicable.

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
