# Peer review of "Key Challenges and Emerging Technologies in Industrial IoT Architectures: A Review"

_sensors, 2022, doi:10.3390/s22155836_

Round 1

Reviewer 1 Report

The notion of Internet of Things was in fact coined by Kevin Ashton and not by the International Telecommunication Union. The authors are advised to refer to the paper, https://ieeexplore.ieee.org/document/9319033, so as to understand the evolution of IoT.

The notion of IoT should be ellaborated in a much more categorical manner in the Introduction section. Also, there needs to be a logical transition from IoT to IIoT.

The contributions of the manuscript-at-hand should also be delineated in more detail at the end of section, Introduction, perhaps in a sequential manner — as (i), (ii), and (iii) / (a), (b), and (c).

Security, Privacy, and Trust in IIoT goes hand-in-hand. Thus, the authors should bringforth the notion of Trust in section 3.1, Key IIoT Requirements. Authors are advised to refer to manuscripts, https://ieeexplore.ieee.org/document/9773986 and https://ieeexplore.ieee.org/document/9099265, and a couple of more, in this regard. The same must be reflected in Table 3 and Figure 8 too.

The authors should also justify the underlying rationale for selecting literature presented in Table 3. Is it the only literature (which is, in fact, not true)? Is it the state-of-the-art, and if so, what's the authors' definition of the state-of-the-art? Please note that literature should be selected primarily based on some specific Methodology and a reference to the same appears to be missing here.

Finally, a careful Proofreading is indispensable in a bid to enhance the quality of the sentence structure in the manuscript-at-hand.

Reviewer 2 Report

Great contribution and this manuscript for sure will be a good reference.

Give space from figures' descriptions and texts.

Conclusion and Future Works need to be improved. 

You could insert more papers from Sensors Journal.

Consider this reference: https://ieeexplore.ieee.org/abstract/document/9514921

Check other important references from Sensors and IEEE Internet of Things

Reviewer 3 Report

The key contribution claimed by the authors of this manuscript:

“In this paper, we first review and compare some widely accepted IIoT reference architectures and present a state-of-the-art review of conceptual and experimental IIoT architectures in literature. We highlight scalability, interoperability, security, privacy, reliability, and low latency as the main IIoT architectural requirements and compare how the current architectures address these challenges. We also highlight the role of emerging technologies in current IIoT architectures to address these requirements and present the literature gap for future research work to address the challenges.”

The English writing, organization, and presentation quality of this work are good. Abstract and Discussion provide a very clear idea about this work and the authors’ contributions. It is a really interesting research survey, containing very updated reference literature. I have noticed a few issues which authors must address to improve the quality of this work.

1.     In the Introduction, it is highly recommended to provide a Table comparing your work with existing reviews or surveys. From the Tables, readers can fastly know the contributions of each study.

2.     Some added Tables are not comprehensive. Authors should check some published surveys and follow Tables presentation with significant details.

3.     Figures are nicely labeled and provide complete information except Figure 1. Authors are suggested to add more discussion about it.

4.     Line 37, there is some writing issue, please make the relevant correction.

5.     Line 54, what is RAMI and IIRA? Authors should carefully check each abbreviation and define at the first place of appearance to enhance better readability.

6.     Overall, Introduction and second section are nicely presented with sufficient reference literature.

7.     Section 3.3.4 and 3.3.6, authors should provide more sufficient discussion and reference literature.

8.     Why did you select architectures from 2015 to 2022?

9.     Tables 3 and 4 are nicely presented. Readers can easily understand the technologies addressed in different studies.

10.    Conclusion and future work section are also nicely written.  

11.     References are very updated and taken from very recent years.

Round 2

Reviewer 1 Report

Thank you for addressing the comments. The overall quality of the manuscript-at-hand has considerably improved. However, I would suggest an appropriate reference to be inserted at Line 19 since Reference [2] is not only pretty old but also doesn't include the facts about Kevin Ashton pioneering the notion of IoT.

On Lines 473 - 474, the authors delineated that SDN facilitates in making Static Networks agile. Although SDN has been conventionally employed for the Static Networks, however, it has been lately employed for the Dynamic Networks too and a lot of Literatrure is prevalent. There is a need to neutralize this argument.

Author Response

Reviewer 1  

Comment 1: 

Thank you for addressing the comments. The overall quality of the manuscript-at-hand has considerably improved. However, I would suggest an appropriate reference to be inserted at Line 19 since Reference [2] is not only pretty old but also doesn't include the facts about Kevin Ashton pioneering the notion of IoT. 

Response: 

The highlighted work in the comment is improved by changing the reference from an old technical report to the website of the International Telecommunication Unit (ITU), which is again the primary source of the mentioned concept. Furthermore, a proper reference is also added to highlight the notion of the Internet of Things concept given by Kevin Ashton.  

Comment 2: 

On Lines 473 - 474, the authors delineated that SDN facilitates in making Static Networks agile. Although SDN has been conventionally employed for the Static Networks, however, it has been lately employed for the Dynamic Networks too and a lot of Literatrure is prevalent. There is a need to neutralize this argument. 

Response: 

We have corrected this by improving the information in the sentence from reference. Following is the updated sentence in the new version of the paper:  

"The Software-Defined Networking (SDN) dynamically manages the distributed network segments to provide optimization and agility in a network with the help of programmable controlling units".